# Gastrectomy for Cancer: A 15-Year Analysis of Real-World Data from the University of Athens

**DOI:** 10.3390/medicina58121792

**Published:** 2022-12-05

**Authors:** Dimitrios Schizas, Konstantinos S. Mylonas, Athanasios Syllaios, Emmanouil I. Kapetanakis, Natasha Hasemaki, Vasileia Ntomi, Adamantios Michalinos, Nikoletta A. Theochari, Christina A. Theochari, Sylvia Krivan, Maria Mpoura, Anargyros Bakopoulos, Ioannis Karavokyros, Theodoros Liakakos

**Affiliations:** 1First Department of Surgery, National and Kapodistrian University of Athens, Laikon General Hospital, 11527 Athens, Greece; 2Third Department of Surgery, National and Kapodistrian University of Athens, Attikon University Hospital, 12462 Athens, Greece; 3Department of Anatomy, European University of Cyprus, Nicosia 2404, Cyprus

**Keywords:** gastrectomy, recurrence, survival, centralization, Greece

## Abstract

*Background and Objectives*: Encouraging data have been reported from referral centers following gastrointestinal cancer surgery. Our goal was to retrospectively review patient outcomes following gastrectomy for gastric or gastroesophageal junction (GEJ) cancer at a high-volume unit of the University of Athens. *Methods*: The enrollment period was from June 2003 to September 2018. Disease-free survival (DFS) and overall survival (OS) were estimated using the Kaplan-Meier method. Cox proportional hazard models were constructed to identify variables independently associated with time-to-event outcomes. *Results*: A total of 205 patients were analyzed. R0 resection was achieved in 183 (89.3%) patients and was more likely to occur following neoadjuvant chemotherapy (*p* = 0.008). Recurrence developed in 46.6% of our cohort and the median disease-free survival was 31.2 months. On multivariate analysis, only staging (HR = 2.15; 95% CI: 1.06–4.36) was independently associated with increased risk of recurrence. All-cause mortality was 57.2% and the median time of death was 40.9 months. On multivariate regression, staging (HR: 1.35; 95% CI: 1.11–1.65) and recurrence (HR: 2.87; 95% CI: 1.32–6.22) predicted inferior prognosis. *Conclusions*: Gastrectomy at the University of Athens has yielded favorable outcomes for patients with GEJ cancer.

## 1. Introduction

Gastric cancer constitutes one of the most prevalent types of visceral malignancies. The incidence of gastric cancer has been estimated at around 950,000 cases per year. Annually, over 700,000 deaths can be attributed to malignancies of the stomach [1]. In recent years, our understanding of the biological processes that mediate the development of gastric cancer has increased [2,3,4]. Of note, mortality as low as 0.3% has been reported for early gastric cancer in Asian countries mainly due to meticulous surveillance programs [5]. In the Western world, however, the prognosis of patients with gastric and gastroesophageal junction (GEJ) carcinomas remains poor, with most survivors developing a locoregional relapse within 24 months of their original treatment [6]. Indeed, major centers in Europe are still reporting five-year survival rates less than 30% [7].

In recent years, the relationship between surgical center volume and patient outcomes has gained significant traction [8]. Admittedly, gastric surgery is technically challenging with an increasing body of literature suggesting that treatment in specialized centers may lead to both superior survival and an increased capacity to address postoperative complications [9,10]. In the Netherlands, after implementation of minimum caseloads and cancer audits, mortality among patients with gastric cancer decreased by half (from 8.0% in 2011 to 4.0% in 2014) [11]. Data from Germany also confirmed that strong centralization reduces in-hospital mortality following complex gastric resections [12].

In the present retrospective study, we explored the outcomes of oncologic gastric surgery over 15 years in a major referral unit of the National and Kapodistrian University of Athens.

## 2. Materials and Methods

### 2.1. Study Design

We retrospectively reviewed the charts of all patients who underwent gastrectomy with curative intent for gastric or gastroesophageal junction cancer at our high-volume upper GI surgical center in Athens, Greece. Our dedicated upper GI team initially operated at Attikon University Hospital but relocated to Laikon General Hospital in 2014. Both institutions are quaternary academic medical centers. The enrollment period was from June 2003 to September 2018. We excluded patients who underwent palliative or open-close procedures. Esophagogastroduodenoscopy (EGD) was utilized to diagnose gastric and GEJ carcinomas. Computed tomography (CT) imaging and laboratory tests with cancer biomarkers were also performed. The 7th edition of the TNM system was used for disease staging [13]. For the first two years after surgery, patients underwent laboratory exams every three months as well as EGD and CT every six months. Subsequently, laboratory testing was performed twice per annum, whereas endoscopy and imaging was repeated on a yearly basis.

The following variables were extracted: demographics, tumor location, histology, grade, stage, surgical approach, extent of resection and lymphadenectomy, postoperative complications, chemotherapy and radiation, recurrence and mortality. Institutional Review Board approval was obtained prior to the start of the study.

### 2.2. Statistical Analysis

The Shapiro–Wilk test was used to determine whether data followed a normal distribution. Normally distributed continuous variables were summarized as means with standard deviations (SDs) whereas non-parametrically distributed data were expressed using medians with interquartile ranges (IQR). Categorical variables were represented with absolute and relative rates. Chi-square was used for hypothesis testing between categorical variables. One-way ANOVA with a Bonferroni correction was used to assess the impact of group membership on continuous variable changes. Multivariate linear and logistic regression models were built using stepwise selection.

Disease-free survival (DFS) and overall survival (OS) were estimated using the Kaplan-Meier method. The log-rank test was used to compare Kaplan-Meier curves. Cox-proportional hazard models were constructed to identify variables independently associated with recurrence and all-cause mortality. Statistical significance was set at *p* < 0.05 for all comparisons and all p-values were two-sided. All statistical analyses were performed in STATA IC15 (StataCorp. College Station, TX: StataCorp LLC).

## 3. Results

### 3.1. Demographics

Overall, a total of 205 patients were eligible for enrollment in our study. Most patients were males (66.8%) with a mean age of 66.2 ± 13.1 years at the time of surgery. The majority of the lesions were gastric carcinomas (73.2%).

### 3.2. Surgical Approach and Postoperative Complications

Overall, the most commonly performed procedure was subtotal gastrectomy (49.2%) followed by total gastrectomy (47.3%). Complete overview of patient clinicopathological features and outcomes stratified by type of gastrectomy and lesion location can be found in Table 1 and Appendix A, respectively. Gastric carcinomas were treated most commonly via subtotal gastrectomy with Roux-en-Y or Billroth II reconstruction (66.7%), whereas the majority of patients with Siewert II (89.3%) and III (92.6%) lesions underwent total gastrectomies with Roux-en-Y reconstruction (*p* < 0.001). R0 resection was achieved in the majority of the patients (89.3%, *p* = 0.02). Neoadjuvant chemotherapy increased the likelihood of R0 resection (*p* = 0.008). Lauren histology (*p* = 0.07), neoadjuvant radiation (*p* = 0.94), and type of surgical procedure (*p* = 0.45) did not affect the radicality of the resection.

Overall, D1 lymphadenectomy was performed in 78.5% of the cases (*p* = 0.01). In the multivariate analysis, there was a higher probability of positive lymph nodes with advanced disease stage (*p* < 0.001) and more extensive lymphadenectomy attempts (*p* < 0.001) (Appendix A). Although 28% of the patients developed postoperative complications, morbidity rates were not affected by tumor location (*p* = 0.15) or procedure type (*p* = 0.75). Controlling for age, tumor location, lymphadenectomy extent, Lauren classification, grade and stage, patients who developed less postoperative complications were more likely to be selected for adjuvant chemotherapy (OR = 0.35, 95% CI: 0.15–0.88), *p* = 0.03 (Appendix A).

### 3.3. Histology, Grade and Stage

According to the Lauren classification, the enteric type (46.9%) was the most prevalent pathology followed by diffuse histology (41.9%); albeit, no statistical significance was reached (p = 0.34). We also found that patients with stage I (22.7%) and II (34.7%) disease usually had enteric type lesions, while those with stage III–IV cancer typically had diffuse type lesions (74.66%), *p* < 0.001 (Table 2).

The majority of tumors were Grade 3 (63.1%), irrespective of locale (*p* = 0.61). Patients with stage I disease typically had grade 1 (45.4%) lesions, stage II was usually associated with grade 2 (38.6%) tumors, whereas most patients suffering from stage III-IV disease had grade 3 (72.3%) carcinomas, *p* < 0.001 (Table 2).

In terms of TNM staging, the majority of lesions were T3 (40.9%) regardless of histological type (*p* = 0.14). Most lesions were N3 (36.7%). Only 5.4% of the patients exhibited metastases at the time of surgery. Tumor locale did not affect the incidence of lymph node (*p* = 0.12) or distant metastases (*p* = 0.33). More than half (52.6%) of the patients had stage III disease. Staging was not significantly influenced by tumor location (*p* = 0.61) or Lauren histology (*p* = 0.05). Staging stratified by tumor location can be found in Appendix A.

### 3.4. Chemotherapy and Radiation

Neoadjuvant chemotherapy was administered in 11.2% of our cohort. Only one patient received radiation prior to undergoing surgery. Adjuvant chemotherapy and radiotherapy rates were 29.7% and 23.4%, respectively. Tumor location and grade did not affect adjuvant chemotherapy or radiotherapy utilization rates. On multivariate analysis, advanced disease stage was associated with increased need for adjuvant chemotherapy (OR: 1.65, 95% CI: 1.21–2.25, *p* = 0.001) and younger patients were likely to receive chemotherapy after surgery. Similarly, adjuvant radiation was inversely related with patient age in the multivariate analysis (OR: 0.95, 95% CI: 0.91–0.99, *p* < 0.003).

### 3.5. Recurrence

Recurrence developed in 46.6% of our cohort and the median disease-free survival was 31.2 months. Tumor location was not associated with relapses (*p* = 0.25, Appendix A). Moving from Attikon University Hospital to Laikon General Hospital in 2014 did not affect recurrence rates (Appendix A). Mixed type lesions (80.0%) recurred more frequently compared to both diffuse (47.6%) and enteric tumors (32.5%) (*p* = 0.04, Figure 1). Grade 3 tumors (80.0%) relapsed at significantly higher rates compared to grade 1 (47.6%) and grade 2 (32.5%) lesions (*p* = 0.01). Overall, on univariate log-rank analysis, the following factors were associated with relapse rates: patient age (*p* < 0.001), Lauren histology (*p* = 0.04), grade (*p* = 0.01), number of positive lymph nodes (*p* < 0.001), and advanced disease stage (*p* < 0.001-Table 2). On multivariate Cox regression analysis, however, only staging (HR = 2.15; 95% CI: 1.06–4.36, *p* = 0.03) remained independently associated with increased risk of recurrence (Table 3).

### 3.6. Survival Analysis

All-cause mortality was 57.2% and the median time of death was 40.9 months. No fatality was noted during the index surgical hospitalization. Survival was comparable among patients with Siewert II (56.2%), Siewert III (63.6%) and frank gastric (56.4%) lesions, *p* = 0.17 (Appendix A). Moving from Attikon University Hospital to Laikon General Hospital in 2014 did not affect survival rates (Appendix A).

Patients with grade 3 (71.3%) lesions had significantly higher all-cause mortality compared to those with grade 1 (0%) and grade 2 (41.0%) tumors (p = 0.001, Figure 2). On univariate analysis, positive lymph nodes (*p* < 0.001), extent of resection (p = 0.02), disease stage (*p* < 0.001-Table 2) and recurrence (*p* < 0.001) were also associated with prognosis.

On multivariate Cox regression, only staging (HR: 1.35; 95% CI: 1.11–1.65, *p* = 0.03) and recurrence (HR: 2.87; 95% CI: 1.32–6.22 *p* = 0.008) were independently associated with increased risk of mortality (Table 4).

## 4. Discussion

A growing body of literature has shown that patients undergoing gastric surgery for cancer at specialized centers require less extensive hospitalization, exhibit higher disease-free and overall survival in addition to enjoying superior quality of life postoperatively [10,14,15]. In the present study, we analyzed the institutional registry of the largest upper GI unit in Athens, the capital of Greece, which accounts for nearly 40% of the national population [16].

During the first decade of our study, our group averaged 10 gastrectomies per year. Importantly, with the progression of time, our annual case volume increased substantially, culminating to 25 patients/year during the last four years of our analysis. Similar development has been observed in other European centers as well. In 2003, Denmark centralized the delivery of gastric resections at initially 37, and ultimately five highly specialized academic units. This led to a staggering decline in postoperative mortality from 8% to 2% [17]. In the Netherlands, a minimum volume of 10 annual gastrectomies per hospital was established in 2012 and increased to 20 operations in 2013. Interestingly, within three years of this paradigm shift, in-hospital mortality following gastric cancer surgery decreased from 8% to 4% [11]. Similarly, recent data from Germany have confirmed that patients treated at dedicated upper GI centers tend to have superior prognosis, at least partly due to the fact that such centers have the resources and expertise to successfully tackle significant postoperative complications [12,18].

The clinical and histological features of our cohort are consistent with published literature. Specifically, the majority of our patients were males in their sixties with T3N3M0, grade 3, enteric type gastric carcinomas. In our series, the most commonly performed procedure was the subtotal gastrectomy. In line with textbook data, patients presenting with stage III–IV disease were more likely to have grade 3, diffuse histology [19,20]. The extent of lymphadenectomy is considered a metric of surgical quality and carries prognostic value. In our cohort, an average of 29.3 lymph nodes were dissected per gastrectomy. This is substantially higher compared to the minimum of 15 lymph nodes, which is necessary for proper staging and prognosis estimation [17,21]. That said, neoadjuvant chemotherapy was only used in approximately 11% of our patients. Originally, disagreements within our multidisciplinary tumor board led to underutilization of neoadjuvant chemotherapy, but in recent years we have been using it increasingly more often for patients with node-positive lesions or those greater than T2.

In our series, the median DFS and overall survival were 31.2 and 40.9 months, respectively, which is comparable to the average prognosis reported from other major European centers [22,23]. Arguably, if neoadjuvant chemotherapy was administered more liberally, we might have noted even more favorable survival rates. Indeed, a compelling body of literature has shown that neoadjuvant chemotherapy improves prognosis not only by increasing the likelihood of R0 excision but also via facilitating histopathological tumor regression, disease downstaging, and eradication of micro-metastasis [24,25,26]. Although, on univariate analysis, patients with grade 3 lesions and higher rates of positive lymph nodes exhibited increased mortality, the impact of histology and lymphatic spread on prognosis was not statistically significant after adjusting for confounders. Only advanced disease stage and recurrence predicted worse survival (again in line with literature) [27]. Notably, the radicality of our resection was independent of tumor location, Lauren histology, and surgical approach. Only neoadjuvant chemotherapy increased the likelihood of R0 resection (which was achieved in nearly 90% of the patients). This figure is notably higher compared to the approximately 65–80% microscopically margin-negative resection rate reported by other institutions [28]. We feel that the aforementioned center-to-center variation is indicative of the importance of identifying the most appropriate candidates for gastrectomy and having them undergo surgery at highly specialized units by expert providers.

Although we practice in a high-volume center, gastrectomy for cancer has not been formally centralized in Greece. This process could be streamlined by following simple steps that have been successfully implemented by other European countries. First, specific centers should receive accreditation to perform gastric resections for malignancies based on internationally accepted criteria for surgical excellence [17]. Second, a national database could be established to monitor the care of patients with gastric and GEJ carcinomas. All Greek citizens are assigned a unique 11-digit civil registration number which could be used to enroll patients to the gastric cancer registry. In Denmark, a national database provides feedback on a three-monthly basis to all of the departments accredited for upper GI surgery in order to enable local audits. Furthermore, a nationwide audit is held annually resulting in an annual report, presenting the national and department specific data along with feedback, summary outcomes and suggestions from a multidisciplinary committee including surgeons, pathologists, oncologists and statisticians [17]. The aforementioned report is distributed to the departments, to all relevant scientific societies and to administrators and is also made publicly available by publication on the internet [17].

The strength of our work lies in the completeness of our dataset. No surgically treated patients are missing from our analysis. Furthermore, it should be emphasized that no operable patients were denied surgery due to resource limitations or other factors [16]. The present work also has certain limitations. First, our study is a retrospective analysis, with a relatively small patient sample within a long-time interval and, therefore, is subject to selection bias. Second, the exact number of referrals during the study period and the time from referral until the start of treatment are unknown. Third, we did not collect data regarding operative approach herein. However, we estimate that approximately 10% of the cases were in fact performed laparoscopically.

All patients were individually assessed by our multidisciplinary tumor board to determine the need for neoadjuvant and/or adjuvant chemotherapy/radiation. Nevertheless, data regarding the utilization rates of each medical regimen were not retrievable from available patient records. That said, during the early years of the study, most patients were treated with the ECF regimen of the MAGIC trial (epirubicin, cisplatin and infused fluorouracil) [29]. In subsequent years, capecitabine and oxaliplatin were shown to be equally as effective as fluorouracil and cisplatin, respectively. It was also conveyed that the addition of docetaxel to CF significantly improved survival and quality of life. The DCF regimen (docetaxel, cisplatin and 5-FU), therefore, became a commonly selected regimen in our center. In light of paradigm-shifting recent data, we currently favor fluorouracil plus leucovorin, oxaliplatin and docetaxel (FLOT4) and oxaliplatin, fluorouracil and folinic acid (FOLFOX) [30,31].

In addition, although gastric carcinomas are notorious for spreading to the liver, lung, and peritoneum, we did not collect granular data regarding distant metastasis sites [32]. Lastly, to our knowledge, our registry is the largest gastrectomy database in Greece and no similar, complete national dataset exists. As such we could not compare our institutional experience with that of another national center.

## 5. Conclusions

We analyzed the institutional database of a quaternary upper GI center in Athens during a 15-year period. In our series, the most commonly performed procedure was subtotal gastrectomy with D1 lymphadenectomy. R0 resection was achieved in nearly 90% of the patients and was more likely to occur following neoadjuvant chemotherapy. Patients presenting with stage III–IV disease were more likely to have grade 3, diffuse histology. Recurrence developed in 46.6% of our cohort and the median disease-free survival was 31.2 months. All-cause mortality was 57.2% and the median time of death was 40.9 months with advanced staging and relapse predicting inferior prognosis.

## Figures and Tables

**Figure 1 medicina-58-01792-f001:**
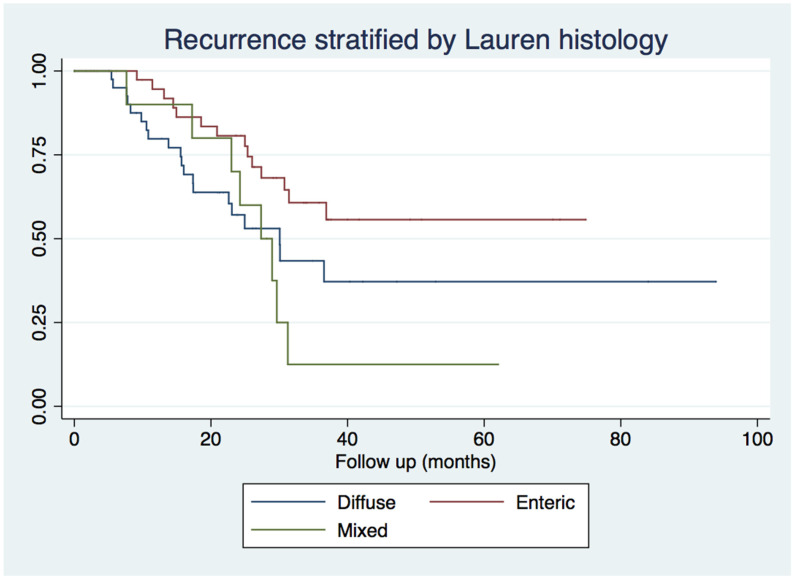
Kaplan Meier curves for recurrence stratified by Lauren histology.

**Figure 2 medicina-58-01792-f002:**
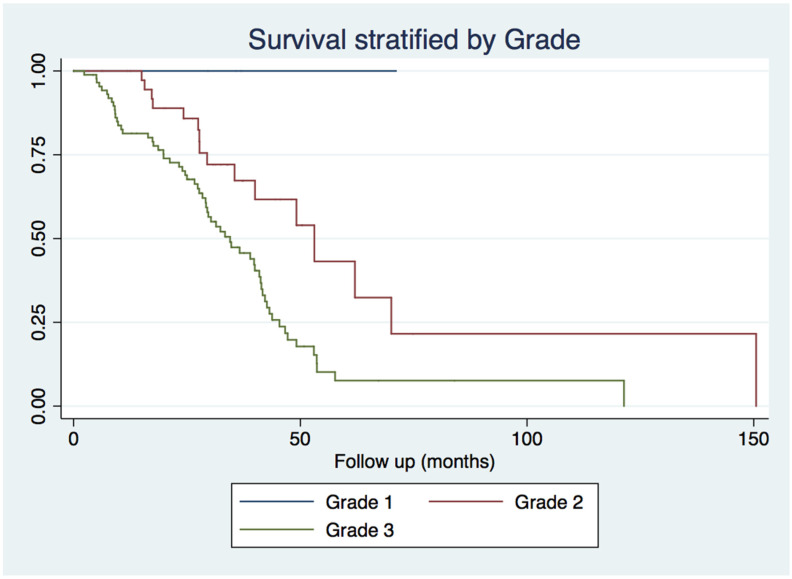
Kaplan Meier curves for overall survival stratified by grade.

**Table 1 medicina-58-01792-t001:** Patient clinicopathological features and outcomes stratified by type of gastrectomy.

Variable	Total Gastrectomy (n = 97; 47.3%)	Subtotal Gastrectomy (n = 101; 49.2%)	Central Gastrectomy (n = 6, 3%)	Wedge Resection (n = 1; 0.5%)	Total (n = 205)	*p*-Value
**Age (y),** *Mean* ± *SD*	65.5 ± 13.6	66.3 ± 12.7	73.8 ± 7.4	76	66.2 ± 13.1	0.27
**Sex**						0.54
*Male*	64 (65.9%)	69 (68.3%)	4 (66.7%)	0 (0%)	137 (66.8%)	
*Female*	33(34.1%)	32 (31.6%)	2 (33.3%)	1 (100%)	68 (33.2%)	
**Location**						** *<0.001* **
*Siewert II*	25 (25.7%)	0 (0%)	3 (50%)	0 (0%)	28 (13.6%)	
*Siewert III*	25 (25.7%)	0 (0%)	2 (33.3%)	0 (0%)	27 (13.1%)	
*Gastric cancer*	47 (48.4%)	101 (100%)	1 (16.7%)	1 (100%)	150 (73.3%)	
**Morbidity**						0.73
*Yes*	28 (31.2%)	20 (25.4%)	1 (20%)	0 (0%)	49 (28.0%)	
*No*	62 (68.8%)	59 (74.6%)	4 (80%)	1 (100%)	126 (72.0%)	
**Neoadjuvant**						
*Chemotherapy*	17 (18.1%)	6 (5.9%)	0 (0%)	0 (0%)	23 (11.2%)	** *0.045* **
*Radiation*	0 (0%)	1 (1%)	0 (0%)	0 (0%)	1 (0.5%)	0.79
Adjuvant						
*Chemotherapy*	69 (74.2%)	68 (67.3%)	3 (50%)	0 (0%)	61 (29.7%)	0.20
*Radiation*	23 (23.9%)	23 (23.2%)	2 (33.3%)	0 (0%)	48 (23.4%)	0.88
**Resection**						0.15
*R0*	80 (82.5%)	96 (95.0%)	6 (100%)	1 (100%)	183 (89.3%)	
*R1*	16 (16.5%)	5 (4.9%)	0 (0%)	0 (0%)	21 (10.2%)	
*R2*	1 (1%)	0 (0%)	0 (0%)	0 (0%)	1 (0.5%)	
**Lymphadenectomy**						0.14
*D0 lymphadenectomy*	0 (0%)	0 (0%)	0 (0%)	0 (0%)	0 (0%)	
*D1 lymphadenectomy*	70 (72.1%)	86 (85.2%)	5 (83.3%)	0 (0%)	161 (78.5%)	
*D2 lymphadenectomy*	24 (24.6%)	15 (14.8%)	1 (16.7%)	0 (0%)	40 (20.0%)	
*D3 lymphadenectomy*	3 (3.3%)	0 (0%)	0 (0%)	0 (0%)	3 (1.5%)	
**Lymph nodes dissected,** *Mean* ± *SD*	33.9 ± 16.4	25.8 ± 16.0	16.6 ± 10.9	3	29.3 ± 16.7	0.53
**Positive lymph nodes,** *Mean* ± *SD*	10.5 ± 13.1	6.5 ± 11.0	6.5 ± 11.3	0	8.4 ± 12.1	0.39
**Lauren classification**						0.36
*Diffuse*	32 (47.4%)	35 (39.3%)	0 (0%)	NA	67 (41.9%)	
*Enteric*	29 (43.3%)	43 (48.4%)	3 (75.0%)	NA	75 (46.9%)	
*Mixed*	6 (8.9%)	11 (12.3%)	1 (25.0%)	NA	18 (11.2%)	
Grade						0.15
*Grade 1*	2 (2.3%)	8 (8.6%)	1 (16.7%)	NA	11 (6.0%)	
*Grade 2*	24 (28.2%)	30 (32.2%)	3 (50.0%)	NA	57 (30.9%)	
*Grade 3*	59 (69.4%)	55 (59.1%)	2 (33.3%)	NA	116 (63.1%)	
**Recurrence**						0.09
*Yes*	34 (52.3%)	25 (39.6%)	1 (100%)	0	60 (46.6%)	
*Censored*	31 (47.7%)	28 (60.4%)	0 (0%)	1	69 (53.4%)	
**All-cause mortality**						0.19
*Yes*	44 (61.1%)	39 (54.1%)	1 (50.0%)	0	84 (57.2%)	
*Censored*	28 (38.9%)	33 (45.8%)	1 (50.0%)	1	63 (42.8%)	

**Table 2 medicina-58-01792-t002:** Lauren classification, grade, recurrence and overall survival stratified by disease stage.

	Staging					*p*-Value
	0	I	II	III	IV	
**Lauren**						** *0.001* **
*Diffuse*	0 (0%)	10 (14.9%)	7 (10.4%)	47 (70.1%)	3 (4.5%)	
*Enteric*	1 (1.3%)	17 (22.7%)	26 (34.7%)	27 (36.0%)	4 (5.3%)	
*Mixed*	0 (0%)	2 (11.1%)	4 (22.2%)	11 (61.1%)	1 (0.6%)	
**Grade**						** *0.001* **
*Grade 1*	0 (0%)	5 (45.4%)	2 (18.2%)	3 (27.2%)	1 (9.2%)	
*Grade 2*	0 (0%)	13 (22.8%)	22 (38.6%)	20 (35.1%)	2 (3.5%)	
*Grade 3*	1 (0.1%)	15 (12.9%)	16 (13.7%)	75 (64.6%)	9 (7.7%)	
**Recurrence**	50%	14.2%	23.3%	64.7%	62.5%	** *0.001* **
**All-cause mortality**	50%	20.8%	41.6%	74.3%	80.0%	** *0.001* **

**Table 3 medicina-58-01792-t003:** Multivariate Cox regression analysis for recurrence.

Stage	Hazard Ratio	95% Confidence Interval	*p*-Value
**Age**	0.99	0.96–1.03	0/84
**Lauren**	0.89	0.53–1.49	0.67
**Grade**	2.64	0.93–7.49	0.06
**Positive lymph nodes**	1.01	0.99–1.04	0.41
**Adjuvant chemotherapy**	0.48	0.15–1.50	0.21
**Adjuvant radiotherapy**	1.20	0.38–1.50	0.21
**Stage**	2.15	1.06–4.36	** *0.03* **

**Table 4 medicina-58-01792-t004:** Multivariate Cox regression analysis for all-cause mortality.

Stage	Hazard Ratio	95% Confidence Interval	*p*-Value
**Grade**	1.16	0.59–2.29	0.66
**Resection extent**	0.86	0.40–1.84	0.70
**Positive lymph nodes**	0.98	0.95–1.01	0.21
**Stage**	1.35	1.11–1.65	** *0.03* **
**Recurrence**	2.87	1.32–6.22	** *0.008* **

## Data Availability

Available by the corresponding author upon request.

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
