# Peer review of "Gastrectomy for Cancer: A 15-Year Analysis of Real-World Data from the University of Athens"

_medicina, 2022, doi:10.3390/medicina58121792_

Round 1
Reviewer 1 Report
Dear Editor, thank you so much for inviting me to revise this manuscript about gastric cancer.
This study addresses a current topic.
The manuscript is quite well written and organized. English could be improved.
Figures and tables are comprehensive and clear.
The introduction explains in a clear and coherent manner the background of this study.
We suggest the following modifications:
· Introduction section: although the authors correctly included important papers in this setting, we believe the changing, evolving systemic treatment scenario for gastric cancer should be further discussed, and some recent studies should be cited within the introduction ( PMID: 33916915; PMID: 33508962; PMID: 33916206 ), only for a matter of consistency. We think it might be useful to introduce the topic of this interesting study.
· Methods and Statistical Analysis: nothing to add.
· Discussion section: Very interesting and timely discussion. Of note, the authors should expand the Discussion section, including a more personal perspective to reflect on. For example, they could answer the following questions – in order to facilitate the understanding of this complex topic to readers: what potential does this study hold? What are the knowledge gaps and how do researchers tackle them? How do you see this area unfolding in the next 5 years? We think it would be extremely interesting for the readers.
However, we think the authors should be acknowledged for their work. In fact, they correctly addressed an important topic, the methods sound good and their discussion is well balanced.
One additional little flaw: the authors could better explain the limitations of their work, in the last part of the Discussion.
We believe this article is suitable for publication in the journal although major revisions are needed. The main strengths of this paper are that it addresses an interesting and very timely question and provides a clear answer, with some limitations.
We suggest a linguistic revision and the addition of some references for a matter of consistency. Moreover, the authors should better clarify some points.
Author Response
November 12th, 2022
Athens, Greece
To the Editorial Board of “Medicina”
We are submitting for consideration our revised manuscript (medicina-1914558) entitled:
«Gastrectomy for Cancer: A 15-year Analysis of Real-World Data from the University of Athens”. Thank you for taking the time and effort to assess our submission so meticulously. Our research group took into account all of your recommendations and we modified our paper accordingly. All manuscript changes have been highlighted with red color. Detailed replies to your comments are provided below:
Reviewer#1
Overall impression: “Dear Editor, thank you so much for inviting me to revise this manuscript about gastric cancer. This study addresses a current topic. The manuscript is quite well written and organized. English could be improved. Figures and tables are comprehensive and clear. The introduction explains in a clear and coherent manner the background of this study…We believe this article is suitable for publication in the journal although major revisions are needed. The main strengths of this paper are that it addresses an interesting and very timely question and provides a clear answer, with some limitations. We suggest a linguistic revision and the addition of some references for a matter of consistency. Moreover, the authors should better clarify some points.”
Authors’ reply: Thank you for appreciating our work and for suggesting ways to further improve it. Please note that our manuscript has been proofed for potential errors in grammar and syntax.
Comment 1.1: “Introduction section: although the authors correctly included important papers in this setting, we believe the changing, evolving systemic treatment scenario for gastric cancer should be further discussed, and some recent studies should be cited within the introduction (PMID: 33916915; PMID: 33508962; PMID: 33916206), only for a matter of consistency. We think it might be useful to introduce the topic of this interesting study.”
Authors’ reply: Thank you for your suggestion. All three studies have been cited as follows (lines 52-53): “In recent years, our understanding of the biological processes that mediate the development of gastric cancer has increased. (2-4)”
Comment 1.2: “Discussion section: Very interesting and timely discussion. Of note, the authors should expand the Discussion section, including a more personal perspective to reflect on. For example, they could answer the following questions – in order to facilitate the understanding of this complex topic to readers: what potential does this study hold? What are the knowledge gaps and how do researchers tackle them? How do you see this area unfolding in the next 5 years? We think it would be extremely interesting for the readers.”
Authors’ reply: Thank you for your questions. We believe these are answered in lines 228-241:
“Although we practice in a high-volume center, gastrectomy for cancer has not been formally centralized in Greece. This process could be streamlined by following simple steps that have been successfully implemented by other European countries. First, specific centers should receive accreditation to perform gastric resections for malignancies based on internationally accepted criteria for surgical excellence.(16) Second, a national database could be established to monitor the care of patients with gastric and GEJ carcinomas. All Greek citizens are assigned a unique 11-digit civil registration number which could be used to enroll patients to the gastric cancer registry. In Denmark, a national database provides feed-back on three-month basis to all of the departments accredited for upper GI surgery in order to enable local audits. Furthermore, a nationwide audit is held annually resulting in an annual report, presenting the national and department specific data along with feedback, summary outcomes and suggestions from a multidisciplinary committee including surgeons, pathologists, oncologists and statisticians.(16) The aforementioned report is distributed to the departments, to all relevant scientific societies and to administrators and is also made publicly available by publication on the internet.(16)”
Comment 1.3: “One additional little flaw: the authors could better explain the limitations of their work, in the last part of the Discussion.”
Authors’ reply: Thank you for your insightful remark. As mentioned in our revised Limitations section: (lines 244-263): “The present work also has certain limitations. First, our study is a retrospective analysis, with a relatively small patient sample within a long-time interval and therefore is subject to selection bias. Second, the exact number of referrals during the study period and the time from referral until the start of treatment are unknown. Third, all patients were individually assessed by our multidisciplinary tumor board to determine the need for neoadjuvant and/or adjuvant chemotherapy/radiation. Data regarding the utilization rates of each medical regimen were not retrievable from available patient records. That said, during the early years of the study most patients were treated with the ECF regimen of the MAGIC trial (epirubicin, cisplatin, and infused fluorouracil).(26) In subsequent recent years, capecitabine and oxaliplatin were shown to be equally effective as fluorouracil and cisplatin respectively. It was also conveyed that the addition of docetaxel to CF significantly improved survival and quality of life. The DCF regimen (docetaxel, cisplatin and 5-FU) therefore became a commonly selected regimen in our center. In light of paradigm-shifting recent data, we currently favor fluorouracil plus leucovorin, oxaliplatin and docetaxel (FLOT4) and oxaliplatin, fluorouracil, and folinic acid (FOLFOX).(27, 28) Fourth, although gastric carcinomas are notorious for spreading to the liver, lungs, and peritoneum, we did not collect granular data regarding distant metastasis sites.(29) Lastly, to our knowledge, our registry is the largest gastrectomy database in Greece and no similar, complete national datasets exists. As such we could not compare our institutional experience with that of another national center.”
Reviewer#2
Overall impression: “This study was a single-center, retrospective study to review patient outcomes following gastrectomy for gastric or gastroesophageal junction (GEJ) cancer at a high-volume unit of the University of Athens. This paper is quite interesting and well written. In addition, large-scaled data from Greek seems to be highly valuable. However, there are several flaws and limitations as follows.”
Authors’ reply: Thank you for appreciating our work and for suggesting ways to further improve it.
Comment 2.1: “The authors need to explain about how to diagnose and determinate the stage for gastric or GEJ cancer in Methods section. Which classification was applied for staging?”
Authors’ reply: Thank you for your questions. As described in lines 77-79: “Esophagogastroduodenoscopy (EGD) was utilized to diagnose gastric and GEJ carcinomas. Computed tomography (CT) imaging and laboratory tests with cancer biomarkers were also performed. The 7th edition of the TNM system was used for disease staging.”
Comment 2.2: “The details in surgical procedures, including procedures, operators, and approaches should be described.”
Authors’ reply: Thank you for your questions. As described in lines 114-117: “Gastric carcinomas were treated most commonly via subtotal gastrectomy with Roux-en-Y or Billroth II reconstruction (66.7%) whereas the majority of patients with Siewert II (89.3%) and III (92.6%) lesions underwent total gastrectomies with Roux-en-Y reconstruction (p<0.001).” Additional details regarding surgical procedures are provided in Table 1 and Supplemental Table 1.
Comment 2.3.1: “The information of indications and regimens of neoadjuvant and adjuvant chemotherapy, as well as radiotherapy, are lack. In addition, the follow-up protocol is also necessary to describe.”
Authors’ reply: Thank you for your suggestions. As described in lines 248-263: “Third, all patients were individually assessed by our multidisciplinary tumor board to determine the need for neoadjuvant and/or adjuvant chemotherapy/radiation. Data regarding the utilization rates of each medical regimen were not retrievable from available patient records. That said, during the early years of the study most patients were treated with the ECF regimen of the MAGIC trial (epirubicin, cisplatin, and infused fluorouracil).(26) In subsequent recent years, capecitabine and oxaliplatin were shown to be equally effective as fluorouracil and cisplatin respectively. It was also conveyed that the addition of docetaxel to CF significantly improved survival and quality of life. The DCF regimen (docetaxel, cisplatin and 5-FU) therefore became a commonly selected regimen in our center. In light of paradigm-shifting recent data, we currently favor fluorouracil plus leucovorin, oxaliplatin and docetaxel (FLOT4) and oxaliplatin, fluorouracil, and folinic acid (FOLFOX).(27, 28) Lastly, to our knowledge, our registry is the largest gastrectomy database in Greece and no similar, complete national datasets exists. As such we could not compare our institutional experience with that of another national center.”
Comment 2.3.2: “In addition, the follow-up protocol is also necessary to describe.”
Authors’ reply: Thank you for your remark. As described in lines 79-82: “For the first two years after surgery, patients underwent laboratory exams every three months as well as EGD and CT every six months. Subsequently, laboratory testing was performed twice per annum, whereas endoscopy and imaging was repeated on a yearly basis.”
Comment 2.4: “This study period was about 15 years, indicating so long. Therefore, chronological bias cannot exclude. Were there differences in outcomes between former and latter terms?”
Authors’ reply: Thank you for your important observation. As described in lines 160-161: “Moving from Attikon University Hospital to Laikon General Hospital in 2014 did not affect recurrence rates (Supplemental Figure 2).”
Furthermore, according to lines 174-175: Moving from Attikon University Hospital to Laikon General Hospital in 2014 did not affect survival rates (Supplemental Figure 4).
Comment 2.5: “Surgical hospital mortality was desirable to indicate.”
Authors’ reply: Thank you for your kind suggestion. As described in lines 171-172: “No fatality was noted during the index surgical hospitalization.”
Comment 2.6: “Recurrence sites are desirable to indicate”
Authors’ reply: Thank you for your remark. As delineated in our Limitations section (lines 259-260): “Fourth, although gastric carcinomas are notorious for spreading to the liver, lungs, and peritoneum, we did not collect granular data regarding distant metastasis sites.(29)
Comment 2.7: “The probabilities of OS and DFS for each pathological stage are desired to indicate”
Authors’ reply: Thank you for your recommendation. These additional data are now illustrated in our revised Table 2
|
|
Staging |
p-value |
||||
|
0
|
I |
II |
III |
IV |
0.001 |
|
|
Lauren |
|
|||||
|
Diffuse |
0 (0%) |
10 (14.9%) |
7 (10.4%) |
47 (70.1%) |
3 (4.5%) |
|
|
Enteric |
1 (1.3%) |
17 (22.7%) |
26 (34.7%) |
27 (36.0%) |
4 (5.3%) |
|
|
Mixed |
0 (0%) |
2 (11.1%) |
4 (22.2%) |
11 (61.1%) |
1 (0.6%) |
|
|
Grade
|
|
0.001 |
||||
|
Grade 1 |
0 (0%) |
5 (45.4%) |
2 (18.2%) |
3 (27.2%) |
1 (9.2%) |
|
|
Grade 2 |
0 (0%) |
13 (22.8%) |
22 (38.6%) |
20 (35.1%) |
2 (3.5%) |
|
|
Grade 3 |
1 (0.1%) |
15 (12.9%) |
16 (13.7%) |
75 (64.6%) |
9 (7.7%) |
|
|
Recurrence |
50% |
14.2% |
23.3% |
64.7% |
62.5% |
0.001 |
|
Overall survival |
50% |
20.8% |
41.6% |
74.3% |
80.0% |
0.001 |
Comment 2.8: “In Discussion section, the authors described that ‘Arguably, if neoadjuvant chemotherapy was administered more liberally, we might have noted even more favorable survival rates.’
However, this speculation is little evident. Because, in the multivariate analyses of this study, R0 resection rate was not included in the independent risk factor for recurrence and all-cause mortality, although neoadjuvant therapy increased the likelihood of R0 resection.”
Authors’ reply: We appreciate your remark. Nevertheless, an overwhelming body of literature has unequivocally shown that neoadjuvant chemotherapy improves outcomes not only by enabling R0 resection in more patients, but also by diminishing the biological burden of disease as the well as the risk of micrometastasis. Therefore, we kindly stand by our comment.
In conclusion, we hope that with these additional revisions, our work is felt appropriate for publication in Medicina. We look forward to hearing from you and we would be pleased to answer any further questions and/or comments you may have.
Sincerely yours,
Dimitrios Schizas, M.D., PhD
Assistant Professor of Surgery
National and Kapodistrian University of Athens
Theodoros Liakakos, M.D., PhD
Professor of Surgery
National and Kapodistrian University of Athens
Reviewer 2 Report
This study was a single-center, retrospective study to review patient outcomes following gastrectomy for gastric or gastroesophageal junction (GEJ) cancer at a high-volume unit of the University of Athens. This paper is quite interesting and well written. In addition, large-scaled data from Greek seems to be highly valuable. However, there are several flaws and limitations as follows.
#1: The authors need to explain about how to diagnose and determinate the stage for gastric or GEJ cancer in Methods section. Which classification was applied for staging?
#2: The details in surgical procedures, including procedures, operators, and approaches should be described.
#3: The information of indications and regimens of neoadjuvant and adjuvant chemotherapy, as well as radiotherapy, are lack. In addition, the follow-up protocol is also necessary to describe.
#4: This study period was about 15 years, indicating so long. Therefore, chronological bias cannot exclude. Were there differences in outcomes between former and latter terms?
#5: Surgical hospital mortality was desirable to indicate.
#6: Recurrence sites are desirable to indicate.
#7: The probabilities of OS and DFS for each pathological stage are desired to indicate.
#8: In Discussion section, the authors described that ‘Arguably, if neoadjuvant chemotherapy was administered more liberally, 194 we might have noted even more favorable survival rates.’ (194-195)
However, this speculation is little evident. Because, in the multivariate analyses of this study, R0 resection rate was not included in the independent risk factor for recurrence and all-cause mortality, although neoadjuvant therapy increased the likelihood of R0 resection.
Author Response

(The authors gave the same response as above.)

Round 2
Reviewer 1 Report
acceptance
Reviewer 2 Report
The revised manuscript was almost appropriately revised. However, a couple of flaws remained.
Comment 2.2: How about approach? The proportions of open and laparoscopic gastrectomy had better been indicated.
Comment 2.7: In the new Table, I'm afraid that 'Overall survival' was incorrect, and maybe had better been replaced into 'mortality'.
In addition, the reviewer could not agree with the response to the comment 2.8.
Comment 2.8: “In Discussion section, the authors described that ‘Arguably, if neoadjuvant chemotherapy was administered more liberally, we might have noted even more favorable survival rates.’
However, this speculation is little evident. Because, in the multivariate analyses of this study, R0 resection rate was not included in the independent risk factor for recurrence and all-cause mortality, although neoadjuvant therapy increased the likelihood of R0 resection.”
Authors’ reply: We appreciate your remark. Nevertheless, an overwhelming body of literature has unequivocally shown that neoadjuvant chemotherapy improves outcomes not only by enabling R0 resection in more patients, but also by diminishing the biological burden of disease as the well as the risk of micrometastasis. Therefore, we kindly stand by our comment.
The reviewer's comment: The basis of this speculation was too weak to discussion. The authors should cite the appropriate references, or should indicate that the R0 resection was the significant risk factor for mortality in the multivariate cox regression analysis in this study.
